# An Osteoblast-Specific Enhancer and Subenhancer Cooperatively Regulate *Runx2* Expression in Chondrocytes

**DOI:** 10.3390/ijms26041653

**Published:** 2025-02-14

**Authors:** Yuki Matsuo, Xin Qin, Takeshi Moriishi, Viviane K. S. Kawata-Matsuura, Hisato Komori, Chiharu Sakane, Suemi Yabuta, Qing Jiang, Hitomi Kaneko, Kosei Ito, Mayo Shigeta, Takaya Abe, Toshihisa Komori

**Affiliations:** 1Department of Molecular Tumor Biology, Nagasaki University Graduate School of Biomedical Sciences, Nagasaki 852-8588, Japan; ysmatsuo@nagasaki-u.ac.jp (Y.M.);; 2Department of Skeletal Development and Regenerative Biology, Nagasaki University Graduate School of Biomedical Sciences, Nagasaki 852-8588, Japan; 3Institute of Orthopaedics, Suzhou Medical College, Soochow University, Suzhou 215006, China; 4Research Center for Biomedical Models and Animal Welfare, Nagasaki University Graduate School of Biomedical Sciences, Nagasaki 852-8588, Japan; 5Laboratory for Animal Resources and Genetic Engineering, RIKEN Center for Biosystems Dynamics Research, Kobe 650-0047, Japan

**Keywords:** Runx2, enhancer, transcriptional regulation, chondrocyte, osteoblast

## Abstract

Runx2 is an essential transcription factor for osteoblast differentiation and chondrocyte maturation. The spatiotemporal expression of Runx2 is regulated by enhancers. We previously identified a 1.3 kb osteoblast-specific enhancer; however, mice with this deletion showed no phenotypes. A 0.8 kb conserved region detected near the 1.3 kb enhancer did not exhibit enhancer activity in reporter assays, whereas four tandem repeats of 452 bp (452 × 4) containing the most conserved region of 0.8 kb induced strong reporter activity in chondrocyte cell lines. However, chondrocytes of enhanced green fluorescent protein (EGFP) reporter mice using 452 × 4 did not express EGFP. When 452 × 4 was combined with the 1.3 kb enhancer, hypertrophic chondrocytes highly expressed EGFP. Moreover, the 0.8 kb region combined with the 1.3 kb enhancer induced EGFP expression in prehypertrophic and hypertrophic chondrocytes. The deletion of both the 1.3 kb enhancer and the 0.8 kb conserved region slightly reduced Runx2 expression in the limbs. However, neither homozygous nor heterozygous deletions in the Runx2^+/−^ background showed phenotypes. The 0.8 kb conserved region itself lacked enhancer activity, but when combined with the 1.3 kb enhancer, EGFP expression was induced in chondrocytes with a similar expression pattern to Runx2. Therefore, the 0.8 kb conserved region has a novel function as a subenhancer.

## 1. Introduction

The spatiotemporal control of gene expression is regulated by promoters and enhancers. Enhancers are non-coding regions located outside the promoter region, are bound by transcription factors, and interact with the promoter to regulate transcription [1]. Candidate enhancers active in a given cell type may be identified by histone modifications, including H3K27ac, H3K4me1, and H3K4me2, and histone variants, such as H2A.Z [2]. The spatiotemporal control of transcription by enhancers has been examined in vivo using reporter systems [3].

In *Drosophila*, two or more redundant (or shadow) enhancers, which control the same gene and drive identical or overlapping expression patterns, are present in many genes [4]. Shadow enhancers have been also identified in *Caenorhabditis elegans*, zebrafish, mice, and humans, and are associated with most developmental genes [5]. Shadow enhancers were previously shown to be functionally redundant because the deletion of individual enhancers showed no phenotype, whereas that of pairs of enhancers resulted in discernible phenotypes [3]. Furthermore, shadow enhancers were found to function additively, synergistically, or redundantly [1]. The overlapping activity of shadow enhancers may have important functions, including phenotypic robustness in the face of environmental and genetic variability, buffering upstream noise through the separation of transcription factor inputs at individual enhancers, and generating more precise expression patterns [3,5,6,7,8,9,10].

Runx2 is an essential transcription factor for osteoblast differentiation and chondrocyte maturation [11]. It is expressed in all osteoblast lineage cells. Runx2 induces commitment of multipotent mesenchymal cells to osteoblast lineage cells, proliferation of osteoprogenitors, differentiation of committed osteoblasts to mature osteoblasts, and expression of major bone matrix protein genes [11]. Runx2 is also expressed in chondrocytes, and its expression is up-regulated in prehypertrophic chondrocytes in the growth plate and maintained in hypertrophic chondrocytes [12,13]. Runx2 induces chondrocyte maturation and enhances their proliferation by inducing Indian hedgehog (*Ihh*) in prehypertrophic chondrocytes [11,14]. Bone morphogenetic protein (BMP) and Wnt signaling pathways induce *Runx2* expression and chondrocyte maturation [15,16,17]. Further, Sox9 suppresses *Runx2* expression and prevents chondrocyte maturation through the induction of *Nkx3-2* expression [18,19].

The transcription of Runx2 is regulated by two promoters, P1 and P2, although the P2 promoter has not been well studied [20,21,22,23,24]. Mice with deletion of the P1 promoter showed delayed skeletal development and severe osteopenia, indicating that transcripts from the P1 promoter play important roles in Runx2-dependent skeletal development [21,22,25]. Transcriptional regulation of Runx2 has been examined in the P1 promoter [20,26,27,28,29,30]. However, reporter mice under the control of the P1 promoter failed to recapitulate Runx2 expression in osteoblasts and chondrocytes [23,31].

We previously described a 1.3 kb osteoblast-specific Runx2 enhancer that directed enhanced green fluorescent protein (EGFP) expression specifically to osteoblasts and showed that 343 bp was sufficient for osteoblast-specific expression [23]. The enhancer was activated by Tcf7, Ctnnb1, Mef2c, Smad1, Sox5/6, Dlx5/6, and Sp7, and formed an enhanceosome in the core 89 bp. Mef2c and Dlx5/6 directly bound to the 89 bp, and the other proteins indirectly bound to it via protein–protein interactions [23]. Transcripts from the P2 promoter were detected not only in osteoblasts but also in osteoprogenitors [32]. As the 1.3 kb osteoblast-specific enhancer directed reporter gene expression to osteoblasts but not to osteoprogenitors, the 1.3 kb enhancer is likely to interact with the P1 promoter [23].

As transcripts from the P1 promoter were detected in both osteoblasts and chondrocytes [22,33], we searched the regulatory regions for Runx2 expression in chondrocytes in the vicinity of the P1 promoter. A 0.8 kb region with homologous sequences has been identified between the 1.3 kb enhancer and the P1 promoter in many species (UCSC Genome Browser on Mouse, July 2007, NCBI/mm9, chr17:44984700-44949500). We examined the enhancer activity of the 0.8 kb region. Although it did not exhibit enhancer activity, the osteoblast-specific 1.3 kb enhancer combined with the 0.8 kb region in reporter mice induced reporter gene expression in chondrocytes with a similar expression pattern to Runx2.

## 2. Results

### 2.1. Deletion of the Osteoblast-Specific 1.3 kb Enhancer Showed No Phenotypes

The 1.3 kb enhancer was located 27 kb upstream of the transcription start site of the Runx2 P1 promoter (Figure 1A). We generated 1.3 kb-deleted mice (1.3^−/−^ mice) using ES cells (Appendix A). Body weight was similar between wild-type and 1.3^−/−^ mice of both sexes at 10 weeks of age (Figure 1B). *Runx2* mRNA levels were also similar in 1.3^−/−^ mice and wild-type mice of both sexes (Figure 1C). A micro-CT analysis of the femurs of 10-week-old mice showed that all parameters of trabecular bone, including bone volume, trabecular thickness, trabecular number, and trabecular bone mineral density (BMD), and those of cortical bone, including cortical volume, cortical thickness, periosteal perimeter, endosteal perimeter, and cortical BMD, were similar between wild-type and 1.3^−/−^ mice (Figure 1D–F).

### 2.2. The 0.8 kb Region Showed Enhancer-like Histone Modifications, and Tandem Repeats of the Core Region Induced Luciferase Activity in Chondrocyte Cell Lines

A chromatin immunoprecipitation (ChIP) analysis revealed that the histone modifications of H3K4me2 and H2A.Z were enriched in the 0.8 kb region in ATDC5 cells, which had been induced to differentiate into chondrocytes for 15 days (Figure 2A). Furthermore, the histone modifications of H3K4me1 and H3K27ac were enriched in the 0.8 kb region in primary chondrocytes, which were grown in a micromass culture to induce maturation for 15 days (Figure 2B).

The enhancer activity of the 0.8 kb DNA fragment was examined using a luciferase vector containing a Hsp68 minimal promoter. The 0.8 kb DNA fragment failed to induce luciferase activity, as did the addition of fibroblast growth factor 2 (Fgf2), Fgf18, bone morphogenetic protein 2 (Bmp2), transforming growth factor β (Tgfβ), retinoic acid (RA), Wnt3a, and sonic hedgehog (Shh), whereas the addition of Ihh slightly increased luciferase activity in chondroprogenitor ATDC5 cells (Figure 2C). Therefore, we constructed a luciferase vector with four tandem repeats of 452 bp DNA, which contained the most homologous region in mouse, rat, human, orangutan, dog, horse, opossum, and chicken 0.8 kb regions (Appendix A), using pGL4.23 containing a minimal promoter. The luciferase activity of 452 × 4 was mildly induced by Fgf2 and strongly induced by Tgfβ in chondrogenic SW1353 cells (Figure 2D). The 452 × 4 vector strongly induced luciferase activity in human chondrocyte cell lines, including SW1353, HCS-TG, and OUMS27, but not in the human osteoblast cell line SaOS2 (Figure 2E,F). A luciferase vector containing two tandem repeats of the 343 bp DNA fragment, the core region of the 1.3 kb enhancer, failed to induce luciferase activity in chondrocyte cell lines but induced it in the osteoblast cell line (Figure 2E,F). A luciferase vector containing 452 × 4 and 343 × 2 induced luciferase activity at similar levels to the 452 × 4 vector in chondrocyte cell lines but failed to induce luciferase activity in the osteoblast cell line (Figure 2E,F). Therefore, 452 × 4 inhibited 343 × 2 activity in the osteoblast cell line. These results indicate that the 0.8 kb region is a candidate chondrocyte-specific enhancer.

### 2.3. The 0.8 kb DNA Region and 1.3 kb Enhancer Cooperatively Directed Reporter EGFP Expression to Chondrocytes

We confirmed the specificity of the 1.3 kb enhancer to osteoblasts by crossing 1.3 kb enhancer–*Hsp68* minimal promoter–EGFP transgenic mice with 2.3 kb *Col1a1* promoter–*Hsp68* minimal promoter–tdTomato transgenic mice. In double-transgenic mice at embryonic day (E) 16.5, the expression of EGFP and tdTomato overlapped in the calvaria (Figure 3A–D), mandible (Figure 3E–H), and limb bones (Figure 3I–L), indicating the osteoblast specificity of the 1.3 kb enhancer.

We generated 452 × 4–*Hsp68* minimal promoter–EGFP transgenic mice. EGFP was detected strongly in the brain and muscle, and a few EGFP-positive cells were present in the growth plate of the femurs at E16.5. EGFP expression was not detected in osteoblasts throughout the body or in chondrocytes in vertebrae (Figure 4). We then generated 1.3 kb–452 × 4–*Hsp68* minimal promoter–EGFP transgenic mice. Osteoblasts were positive for EGFP throughout the body, and all hypertrophic chondrocytes were positive for EGFP in vertebrae, whereas a limited number of hypertrophic chondrocytes in the growth plate of femurs were positive for EGFP at E16.5. Furthermore, EGFP expression was detected in muscle but not in the brain (Figure 5). We also examined the 1.3 kb–452 × 4 transgenic mice at E15.5. Most hypertrophic chondrocytes were EGFP-positive in the femurs and vertebrae, in addition to the EGFP expression in osteoblasts throughout the body (Figure 6). Further, we generated 1.3 kb–0.8 kb–*Hsp68* minimal promoter–EGFP transgenic mice. Osteoblasts were positive for EGFP throughout the body. EGFP expression was detected in prehypertrophic and hypertrophic chondrocytes in the growth plate of femurs. These cells were also positive for EGFP in vertebrae. EGFP expression was weakly detected in the brain but not in muscle (Figure 7). We compared the EGFP expression pattern with the Runx2 protein expression pattern in the growth plate of wild-type mice. The expression of the Runx2 protein was low in proliferating chondrocytes and was up-regulated in prehypertrophic chondrocytes and hypertrophic chondrocytes (Appendix A). Therefore, the expression pattern in the growth plate of 1.3 kb–0.8 kb–*Hsp68* minimal promoter–EGFP transgenic mice was similar to that of the Runx2 protein in wild-type mice. These results indicate that the 0.8 kb region is a regulatory element that directs the chondrocyte expression of Runx2 through cooperation with the 1.3 kb enhancer.

### 2.4. The Deletion of 1.3 kb and 0.8 kb Regions Showed No Apparent Phenotypes

We generated mice with the deletion of the 1.3 kb and 0.8 kb regions (1.3;0.8^−/−^) by injecting guide RNA into the fertilized eggs of 1.3^−/−^ mice (Appendix A). *Runx2* mRNA levels in the calvaria were similar in 1.3;0.8^−/−^ mice and wild-type mice, whereas those in the limbs were lower in 1.3;0.8^−/−^ mice than in wild-type mice (Figure 8A).

A histological analysis of femurs exhibited no apparent delay in endochondral ossification, as shown in the similar length of bone marrow in wild-type and 1.3;0.8^−/−^ mice at E15.5 (Figure 8B–D). Further, the length of femurs at postnatal day 1 (P1) were similar between wild-type and 1.3;0.8^−/−^ mice (Figure 8E). Therefore, we generated 1.3;0.8^+/−^/*Runx2*^+/−^ mice by crossing 1.3;0.8^+/−^ mice with *Runx2*^+/−^ mice to observe the functions of the 1.3 kb and 0.8 kb regions in the haplodeficient state of *Runx2*. In comparisons with wild-type mice, the mineralization of calvariae and ribs was retarded and clavicles were hypoplastic at E15.5 and E18.5 in both *Runx2*^+/−^ and 1.3;0.8^+/−^/*Runx2*^+/−^ mice (Figure 8F,G). However, no apparent differences were observed between *Runx2*^+/−^ and 1.3;0.8^+/−^/*Runx2*^+/−^ mice. Further, in comparisons with wild-type mice, a histological analysis at E15.5 showed that the process of endochondral ossification was slightly retarded in *Runx2*^+/−^ and 1.3;0.8^+/−^/*Runx2*^+/−^ mice (Figure 8H–M). Although the levels of the delay were different among individual *Runx2*^+/−^ and 1.3;0.8^+/−^/*Runx2*^+/−^ mice, the delay was similarly observed between *Runx2*^+/−^ and 1.3;0.8^+/−^/*Runx2*^+/−^ mice (Figure 8H–M, Appendix A). Therefore, there was no apparent delay in intramembranous or endochondral ossification in 1.3;0.8^+/−^/*Runx2*^+/−^ mice compared to *Runx2*^+/−^ mice.

## 3. Discussion

Although the histone of the 0.8 kb region was modified similarly to an enhancer (Figure 2A,B), the 0.8 kb DNA fragment did not exhibit enhancer activity in vitro (Figure 2C). However, the four tandem repeats of 452 bp (452 × 4), the core homologous region of 0.8 kb, exhibited enhancer activity specifically in chondrocyte cell lines (Figure 2D–F) but virtually no activity in chondrocytes in vivo (Figure 4). When 452 × 4 was combined with the 1.3 kb osteoblast-specific enhancer in EGFP reporter mice, EGFP expression was detected in osteoblasts and hypertrophic chondrocytes (Figure 5 and Figure 6). Therefore, the 0.8 kb region was not an enhancer, regulating the spatiotemporal expression of the corresponding gene, and appeared to be a subenhancer that does not exhibit sufficient activity as an enhancer but acquires enhancer activity when combined with an enhancer.

In mesoderm development in *Drosophila*, 64% of loci were found to have shadow enhancers, indicating their pervasiveness throughout the genome [9]. Many shadow enhancers have an overlapping spatiotemporal expression pattern and exhibit activity to acquire the necessary level of expression that ensures phenotypic robustness or a faithful expression pattern, or to maintain a low level of expression noise against genetic perturbations and environmental stresses, such as variations in temperature [4,5,6,7,8,9,35]. In mice, *Ihh*, which is required for skeletal development, is regulated by at least nine enhancers [10]. They have individual expression specificities with some overlap in the digit anlagen, growth plates, skull sutures, and fingertips [10]. Sequential deletions of these enhancers resulted in more severe growth defects of the skull and long bones, indicating that these enhancers function in an additive manner [10]. The pervasive presence of multiple enhancers was also observed at seven distinct loci required for limb development [3]. Although the single deletion of each of the ten enhancers, which had individual expression patterns with overlap, did not show phenotypes, the deletion of pairs of limb enhancers near the same gene caused phenotypes, indicating that these enhancers function redundantly in limb development [3]. Furthermore, single enhancer deletions showed phenotypes in heterozygous mutant mice of the target gene, indicating that redundant enhancers function additively [3]. These shadow or redundant enhancers in *Drosophila* and mice have individual expression patterns with overlap and function additively to ensure phenotypic robustness or faithful expression patterns. However, the 0.8 kb region alone directed EGFP expression to neither osteoblasts nor chondrocytes, in which Runx2 is expressed (Figure 4). Furthermore, combined with the 1.3 kb osteoblast-specific enhancer, the 0.8 kb region did not function additively to induce expression in osteoblasts but induced expression in a different lineage of cells, chondrocytes (Figure 5, Figure 6 and Figure 7). Therefore, the 0.8 kb region is not a shadow or redundant enhancer. We propose that the 0.8 kb subenhancer alone cannot interact with the promoter, but the 1.3 kb osteoblast-specific enhancer and the 0.8 kb subenhancer cooperatively interact with the promoter and induce Runx2 transcription in chondrocytes (Figure 9). The 452 × 4 luciferase vector induced activity in chondrocyte cell lines, whereas the 452 × 4 DNA fragment failed to induce EGFP expression in chondrocytes in vivo (Figure 2E and Figure 4). Therefore, it is speculated that the chromatin modifiers in the 1.3 kb osteoblast-specific enhancer complex enhanced chromatin modification in the 0.8 kb subenhancer and/or the 1.3 kb enhancer complex supplied transcription factors and cofactors to the 0.8 kb subenhancer, either of which led to the interaction of three complexes (enhancer, subenhancer, and promoter) in chondrocytes in vivo (Figure 9). As the chromatin of the 0.8 kb region was modified as an enhancer in a chondrogenic cell line and primary chondrocytes (Figure 2A,B), the 0.8 kb region was considered to be activated, as shown in Figure 9C. This is the first study to report a subenhancer and show a new model of cooperation between two regulatory elements, namely, an enhancer and a subenhancer.

Runx2 expression is low in the growth plate of the resting chondrocyte layer, is up-regulated from the proliferating chondrocyte layer to the prehypertrophic chondrocyte layer, and high expression is maintained in the hypertrophic chondrocyte layer (Appendix A). The expression pattern of EGFP in the growth plate of femurs was closer to that of Runx2 in reporter mice using the 1.3 kb enhancer and 0.8 kb region than in those using the 1.3 kb enhancer and 452 × 4 (Figure 5, Figure 6 and Figure 7). Mef2c is required for chondrocyte maturation, and its expression pattern is similar to that of Runx2 in the growth plate [36]. There are two Mef2c-binding motifs in the 0.8 kb region (Appendix A). Since they are located in a less conserved region, this region was not included in 452 bp. Although the 0.8 kb luciferase vector had virtually no reporter activity in vitro, the introduction of *Mef2c* strongly induced its reporter activity in SW1353 cells (Figure 2C, Appendix A). The 1.3 kb enhancer had one conserved Mef2c motif, which was bound by Mef2c, in the 343 bp core region, and Mef2c was required for enhancer activity [23]. Reporter mice using the 343 bp region showed osteoblast-specific expression, and EGFP expression was absent in chondrocytes [23]. However, reporter mice using four tandem repeats of 89 bp, which is the most conserved region in 343 bp and contains the Mef2c motif, showed EGFP expression in both osteoblasts and chondrocytes, and the expression pattern in the growth plate was similar to that of Runx2 [23]. These findings suggest that the presence of multiple Mef2c motifs directed EGFP expression in the growth plate to a similar pattern as that of Runx2, implying an important role for Mef2c in Runx2 expression in the growth plate.

**Figure 9 ijms-26-01653-f009:**
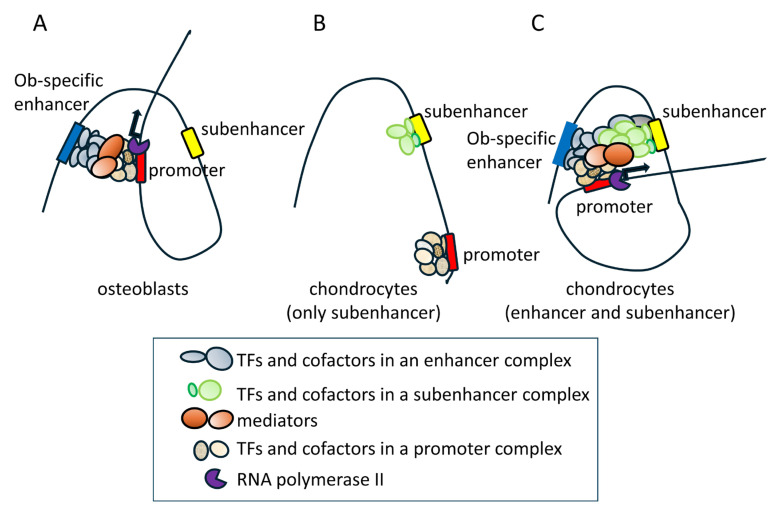
Proposed mechanism of Runx2 transcription in chondrocytes by the enhancer and subenhancer. Complexes composed of transcription factors and cofactors are formed in enhancer, subenhancer, and promoter regions. Some transcription factors bind to the DNA directly, and other transcription factors and cofactors bind to the DNA-bound transcription factors or bind one another by protein–protein interactions, forming a complex. Mediators modulate the interaction of enhancer and promoter complexes and enhance transcription by RNA polymerase II [37]. (**A**) The 1.3 kb osteoblast-specific enhancer interacts with the promoter in osteoblasts and induces Runx2 transcription. Ob: osteoblast. (**B**) The 0.8 kb subenhancer cannot induce Runx2 transcription in chondrocytes in the absence of the 1.3 kb enhancer, probably because the complex formation in the 0.8 kb subenhancer is incomplete for the interaction with the promoter due to insufficient chromatin remodeling and/or a lack of some transcription factors and cofactors in the complex. (**C**) The 1.3 kb enhancer, 0.8 kb subenhancer, and promoter interact and induce Runx2 transcription in chondrocytes. The complexes of the 1.3 kb enhancer and 0.8 kb subenhancer interact with the promoter, probably due to the enhanced chromatin remodeling of the 0.8 kb subenhancer by chromatin modeling factors in the 1.3 kb enhancer complex and/or the supply of the lacking transcription factors and cofactors by the 1.3 kb enhancer complex. Arrows indicate the transcription by RNA polymerase II. TFs: transcription factors.

Although the deletion of the 1.3 kb region or of both the 1.3 kb and 0.8 kb regions showed no apparent phenotype (Figure 1 and Figure 8), *Runx2* expression was slightly reduced in the limbs by the deletion of both the 1.3 kb and 0.8 kb regions (Figure 8A). However, the extent of this reduction was not sufficient to show phenotypes, even in the background of *Runx2* haplodeficiency (Figure 8F–M, Appendix A). Therefore, *Runx2* expression is also regulated by multiple enhancers with redundant functions, similar to other developmental genes.

In conclusion, we herein revealed that cooperation between an osteoblast-specific enhancer and a neighboring subenhancer induced expression in chondrocytes. ChIP- and the assay for transposase-accessible chromatin (ATAC)-sequencing recently revealed many enhancer candidates, the activities and physiological functions of which need to be examined in vivo. However, regions with low or no enhancer activity in vivo might have physiological functions in cooperation with other enhancers. The genomic region examined in this study was restricted to the vicinity of the P1 promoter. As Runx2 is an essential transcription factor for skeletal development, further studies on transcriptional regulation by enhancers distributed in a wide range of the genome are extremely important for obtaining a more detailed understanding of the process of skeletal development.

## 4. Materials and Methods

### 4.1. Generation of Transgenic and Runx2 Enhancer-Deletion Mice

CAG-Flp transgenic mice, CAG-Cre transgenic mice, 1.3 kb enhancer–*Hsp68* minimal promoter–EGFP reporter mice, 2.3 kb *Col1a1* promoter–tdTomato reporter mice, and *Runx2*^+/−^ mice were previously described [23,38,39]. The generation of 1.3 kb-floxed mice (Accession No. CDB1267K: https://large.riken.jp/distribution/mutant-list.html, accessed on 1 December 2020) were generated using HK3i ES cells [40], which were derived from C57BL/6 embryos, by the homologous recombination of the targeting vector carrying a 1.3 kb enhancer flanked by Loxp and a neomycin-resistance gene flanked by Frt. To remove Neo flanked by Frt, offspring (F1) were crossed with CAG-Flp transgenic mice. To generate 1.3^−/−^ mice, 1.3^fl/fl^ mice were mated with CAG-Cre transgenic mice, and 1.3^−/−^ mice were then confirmed by PCR using the primers shown in Appendix A. To obtain the 452 × 4 DNA fragment, four 452 bp DNA fragments were tandemly subcloned into pBluescript II. To generate EGFP reporter mice, 452 × 4, the 1.3 kb DNA fragment and 452 × 4, or the 1.3 kb and 0.8 kb DNA fragments were cloned into pBluescript II with the *Hsp68* minimal promoter, and EGFP reporter mice were generated as previously described [23]. To generate mice with the deletion of both the 1.3 kb and 0.8 kb regions (1.3;0.8^−/−^), two gRNAs that delete the 0.8 kb region and Cas9 RNA were injected into the cytoplasm of fertilized eggs from 1.3^−/−^ mice (Appendix A). PCR using the primers shown in Appendix A confirmed 1.3;0.8^−/−^ mice. All animal studies were performed in the C57BL/6 background. Before the study, all experimental protocols were reviewed and approved by the Animal Care and Use Committee of the Nagasaki University Graduate School of Biomedical Sciences (No. 0906170767) and the Institutional Animal Care and Use Committee of RIKEN Kobe Branch. Mice were reared in a pathogen-free environment on a 12 h light cycle at 22 ± 2 °C with standard chow (CLEA Japan, Tokyo, Japan) and free access to tap water. All relevant guidelines for work with animals were adhered to in this study.

### 4.2. Skeletal Preparation and Micro-CT Analysis

Skeletal preparations were performed as described [39]. Micro-CT analyses were conducted by fixing the femurs of 10-week-old mice in ethanol and quantifying trabecular and cortical bone using a micro-CT system (R_mCT; Rigaku Corporation, Tokyo, Japan). Data from scanned slices were used for a three-dimensional analysis to calculate femoral morphometric parameters. Femoral trabecular bone parameters were measured on the distal femoral metaphysis. Approximately 2.4 mm (0.5 mm from the growth plate) was cranio-caudally scanned, and 200 slices in 12 μm increments were taken. Cortical bone parameters were measured in the mid-diaphysis of femurs.

### 4.3. Cell Lines

The ATDC5 cell line was purchased from the RIKEN Cell Bank (Tsukuba Science City, Tsukuba, Japan). The SW1353 cell line was obtained from ATCC. The SaOS2 cell line was obtained from the Resource Center for Biomedical Research, Institute of Development, Aging and Cancer, Tohoku University. The HCS-TG cell line was obtained from Dr. Hideki Yoshikawa (Osaka University, Suita, Japan) [41]. The OUMS27 cell line was obtained from Dr. Masayoshi Namba (Okayama University, Okayama, Japan) [42].

### 4.4. Cell Culture and ChIP Analysis

ATDC5 cells, which are a mouse chondroprogenitor cell line, were cultured in Dulbecco’s Modified Eagle’s Medium (DMEM)/Ham’s F-12 (1:1) hybrid medium (Thermo Fisher Scientific Inc., Waltham, MA, USA) supplemented with 5% fetal bovine serum (FBS) (Thermo Fisher Scientific Inc.), 10 μg/mL human transferrin (Roche, Basel, Switzerland), and 3 × 10^−8^ M sodium selenite (Merck, Darmstadt, Germany). To induce chondrogenesis, ATDC5 cells were plated at a density of 6 × 10^4^ cells/well in six-well plates and cultured for 15 days in the above medium supplemented with 10 μg/mL of human recombinant insulin (FUJIFILM Wako Pure Chemical, Osaka, Japan). Primary chondrocytes were isolated from the limb skeletons of wild-type embryos at E15.5 and then cultured in DMEM/Nutrient Mixture F-12 Ham (Merck) containing 5% FBS and 10 μg/mL transferrin (Roche). Primary chondrocytes were suspended in DMEM/F12 (Thermo Fisher Scientific Inc.) containing 5% FBS, 10 μg/mL transferrin (Roche), 50 μg/mL ascorbic acid (FUJIFILM Wako Pure Chemical), and 10 mM β-glycerophosphate at a concentration of 1.0 × 10^5^ cells/mL, and two 10 μL drops of cells per grid were placed in a 150 mm gridded culture dish. All chondrogenic cultures were harvested on day 15 and subjected to ChIP analyses. ChIP analyses were performed using rabbit polyclonal anti-H2A.Z (Abcam, Cambridge, UK; ab4174), anti-H3K27ac (Abcam, ab4729), anti-H3K4me1 (Abcam, ab8895), and H3K4me2 (Abcam, ab35356) antibodies and mouse IgG (Cell Signaling, Danvers, MA, USA). PCR was conducted in a 25 μL tube using KOD FX Neo DNA polymerase (Toyobo, Osaka, Japan). The real-time quantitative PCR assay was performed using a Roche LightCycler 480 qPCR machine with SYBR green (Roche). Primer sequences are shown in Appendix A. The human chondrosarcoma (chondrocyte) cell lines SW1353, HCS-TG, and OUMS27 and the human osteosarcoma (osteoblast) cell line SaOS2 were cultured in DMEM (Merck) supplemented with 10% FBS (NICHIREI Biosciences Inc. Tokyo, Japan). All culture media were supplemented with 100 U/mL of penicillin (Nacalai Tesque, Inc. Kyoto, Japan) and 100 μg/mL of streptomycin (Nacalai Tesque, Inc.).

### 4.5. Histological Analyses

Regarding fluorescent signal observations, embryos at E16.5 or E15.5 were fixed with 4% paraformaldehyde/phosphate-buffered saline (PBS) at 4 °C for 2 h, washed with PBS at 4 °C for 1 h, immersed in 20% sucrose at 4 °C overnight, and then embedded in O. C. T. Compound (Sakura Finetek, Tokyo, Japan). Seven-micrometer-thick sections were examined using an All-in-one Fluorescence Microscope (BZ-X700; KEYENCE, Osaka, Japan). Regarding H-E staining, the limbs were fixed with 4% paraformaldehyde/PBS at 4 °C overnight, decalcified in 10% EDTA (pH 7.4), and embedded in paraffin. Three-micrometer-thick sections were examined using Leica CM3050S (Leica Biosystems, Tokyo, Japan).

### 4.6. Reporter Assay

The 343 × 2 DNA fragment was previously described [23]. The 0.8 kb DNA fragment was subcloned into *Hsp68* minimal promoter-pGL4.10. The 452 × 4 DNA fragment and the DNA fragment of 452 × 4 combined with 343 × 2 were subcloned into pGL4.23 (Promega, Madison, WI, USA). ATDC5, SW1353, HCS-TG, and SaOS2 cells were transfected with plasmid DNAs (0.05 μg luciferase reporter vector and 0.05 μg pRL-TK Renilla) using X-tremeGENE9 (Roche). OUMS27 cells were transfected with plasmid DNAs (0.2 μg luciferase reporter vector and 0.2 μg pRL-TK Renilla) using the Neon Transfection System (Thermo Fisher Scientific). Luciferase activities were examined using the Dual-Luciferase Reporter Assay System (Promega) and normalized to *Renilla* luciferase activity.

### 4.7. Real-Time RT-qPCR

Bone marrow was flushed out from tibiae by PBS at 10 weeks of age, and total RNA was extracted from tibiae using ISOGEN (NIPPON GENE, Tokyo, Japan). RNA was also extracted from the calvariae, forelimbs, and hindlimbs of newborn mice. Real-time RT-qPCR was performed using a THUNDERBIRD SYBR qPCR Mix (Toyobo, Osaka, Japan) and Light Cycler 480 real-time PCR system (Roche). Primer sequences are shown in Appendix A. We normalized values to those of *Actb*.

### 4.8. Statistical Analysis

Normality and homogeneity of variance were tested, and statistical analyses of two groups were performed using the Student’s *t*-test and those of more than three groups by a one-way ANOVA using BellCurve for Excel ver. 4.08 (Social Survey Research Information Co., Ltd., Tokyo, Japan). A *p*-value < 0.05 was considered to be significant.

## Figures and Tables

**Figure 1 ijms-26-01653-f001:**
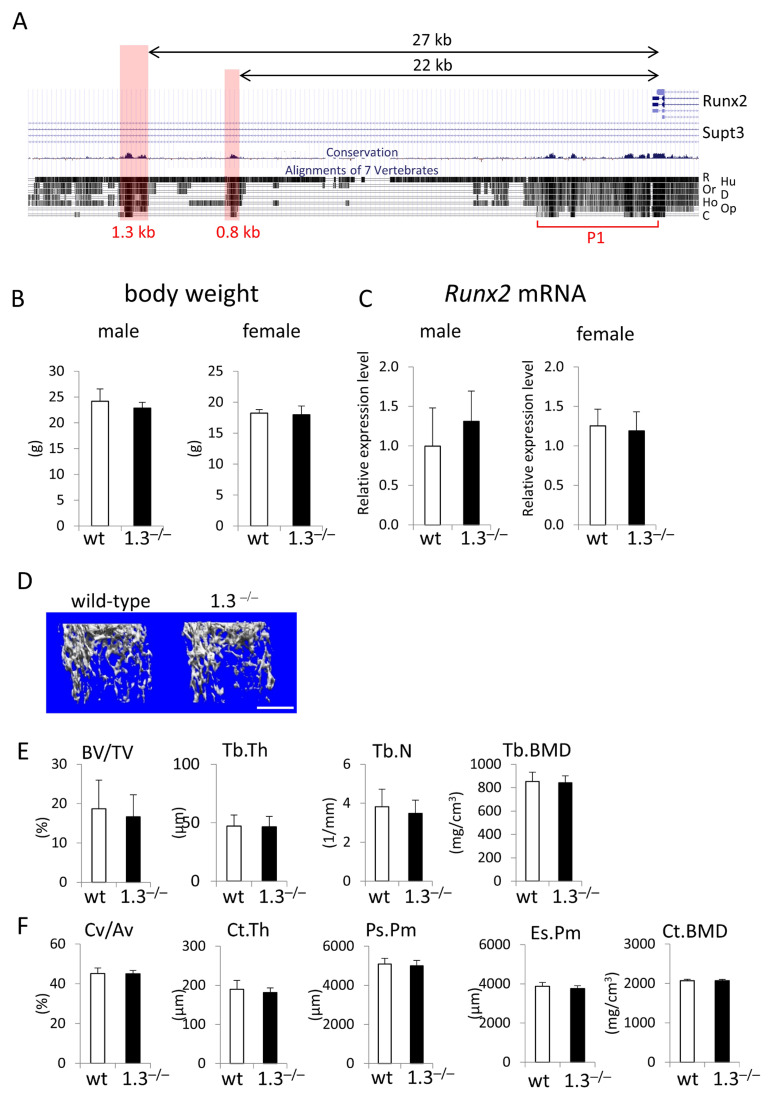
Phenotypes of 1.3 kb enhancer-deletion mice at 10 weeks of age. (**A**) UCSC Genome Browser screenshot showing homology against the mouse sequence in the upstream region of exon 1 of *Runx2*. R: rat, Hu: human, Or: orangutan, D: dog, Ho: horse, Op: opossum, C: chicken. The locations of the 1.3 kb, 0.8 kb, and Runx2 P1 promoter (P1) regions are shown. (**B**) Body weights and (**C**) tibial *Runx2* mRNA expression in wild-type (wt) and 1.3^−/−^ mice. (**D**–**F**) Micro-CT analysis of femurs in male wild-type and 1.3^−/−^ mice. (**D**) Three-dimensional trabecular bone architecture of distal femoral metaphysis. (**E**) Quantification of the trabecular bone volume (bone volume/tissue volume, BV/TV), trabecular thickness (Tb.Th), trabecular number (Tb.N), and trabecular bone mineral density (Tb.BMD). (**F**) Quantification of the cortical bone ratio (cortical bone volume/all bone volume, Cv/Av), cortical thickness (Ct.Th), periosteal perimeter (Ps.Pm), endosteal perimeter (Es.Pm), and cortical bone mineral density (Ct.BMD). The numbers of mice analyzed were as follows: male wild type, n = 6; 1.3^−/−^, n = 7–8; female wild type, n = 7–8; 1.3^−/−^, n = 12–13, in B and C; and wild type, n = 6; 1.3^−/−^, n = 8, in E and F. Scale bar: 1 mm.

**Figure 2 ijms-26-01653-f002:**
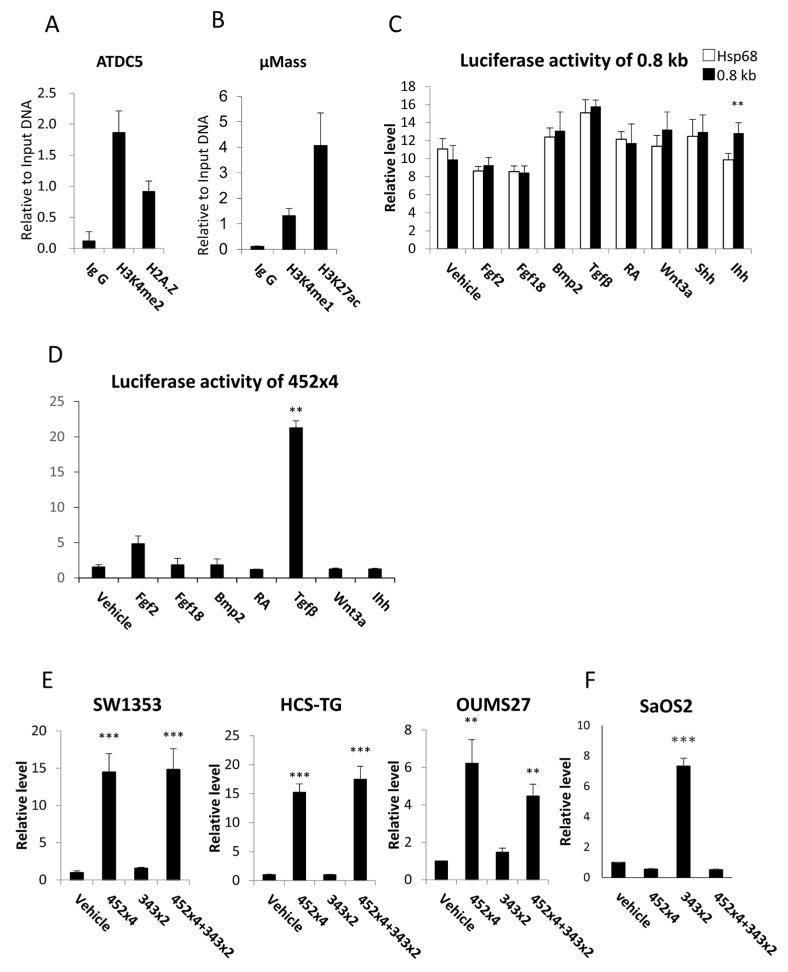
ChIP analysis of the 0.8 kb region and reporter assays of the 0.8 kb DNA and 452 × 4. (**A**,**B**) ChIP analysis of ATDC5 cells (**A**) and a micromass culture of primary chondrocytes (**B**). Antibodies against H3K4me2 and H2A.Z (**A**) and H3K4me1 and H3K27ac (**B**) were used for immunoprecipitation. (**C**–**F**) Reporter assays. Induction of the reporter activity of the 0.8 kb luciferase vector in ATDC5 cells (**C**) and the 454 × 4 luciferase vector in SW1353 cells (**D**) by various factors. RA: Retinoic acid. Reporter activity of the 452 × 4, 343 × 2, and 452 × 4 + 343 × 2 luciferase vectors in chondrocyte cell lines (**E**) and in the osteoblast cell line (**F**). Data are the means ± SEs. Versus *Hsp68* in (**C**) and versus vehicle in (**D**–**F**), ** *p* < 0.01, *** *p* < 0.001. Similar results were obtained from two independent experiments (**C**,**D**) and three to five independent experiments (**E**,**F**), and representative data are shown.

**Figure 3 ijms-26-01653-f003:**
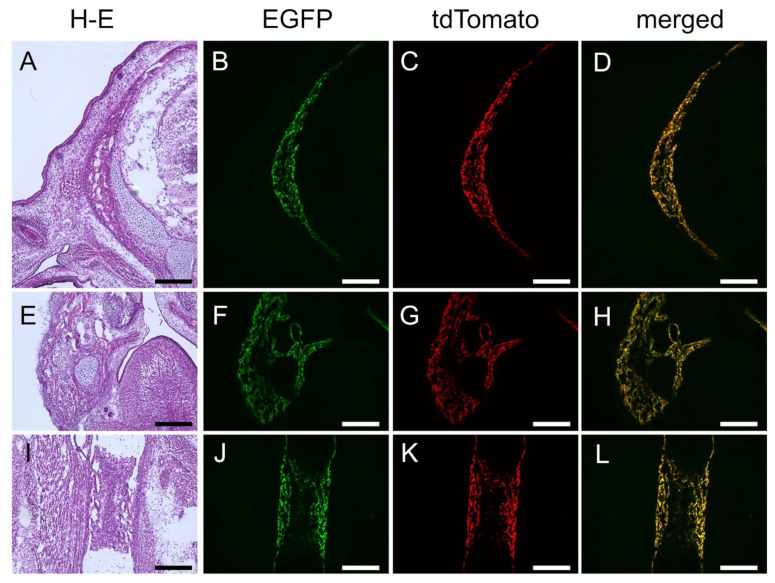
Reporter gene expression in 1.3 kb enhancer–EGFP and Col1a1–tdTomato double-transgenic mice at E16.5. Frozen sections of the head (**A**–**D**), mandible (**E**–**H**), and femur (**I**–**L**). (**A**,**E**,**I**) Hematoxylin and eosin (**H**–**E**) staining. (**B**,**F**,**J**) EGFP expression. (**C**,**G**,**K**) tdTomato expression. (**D**,**H**,**L**) Merged pictures of EGFP and tdTomato expression. Scale bar: 200 μm.

**Figure 4 ijms-26-01653-f004:**
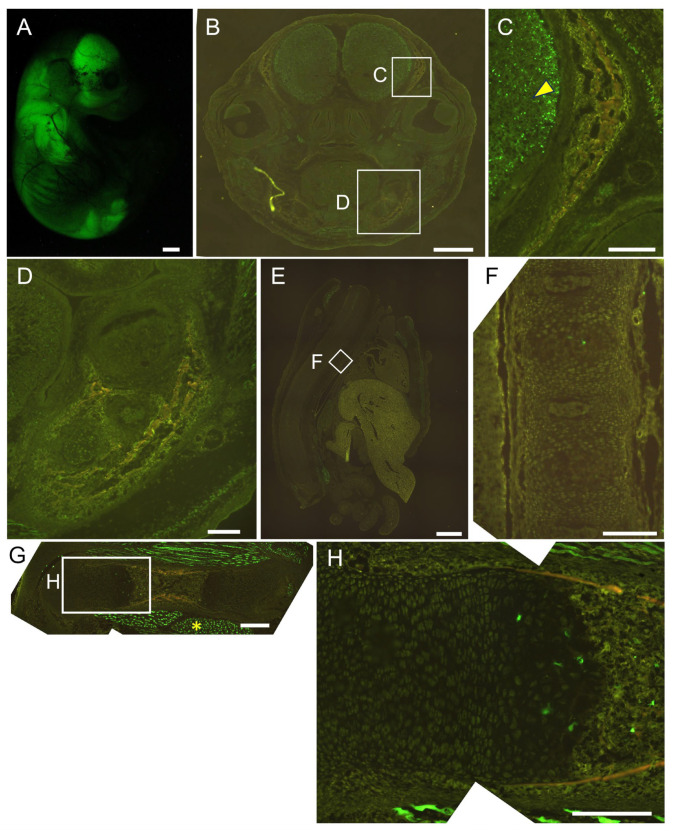
Reporter gene expression of 452 × 4-EGFP mice at E16.5. Appearance (**A**) and frozen sections of the head (**B**,**C**), mandible (**D**), vertebrae (**E**,**F**), and femur (**G**,**H**). The boxed regions in (**B**,**E**,**G**) are magnified in (**C**,**D**,**F**,**H**). Arrowhead in (**C**) shows brain, and asterisk in (**G**) shows muscle. Scale bars: 1 mm (**A**,**B**,**E**), 200 μm (**C**,**D**,**F**–**H**). Three out of three F0 transgenic mice analyzed showed a similar expression pattern, and a representative F0 mouse is shown.

**Figure 5 ijms-26-01653-f005:**
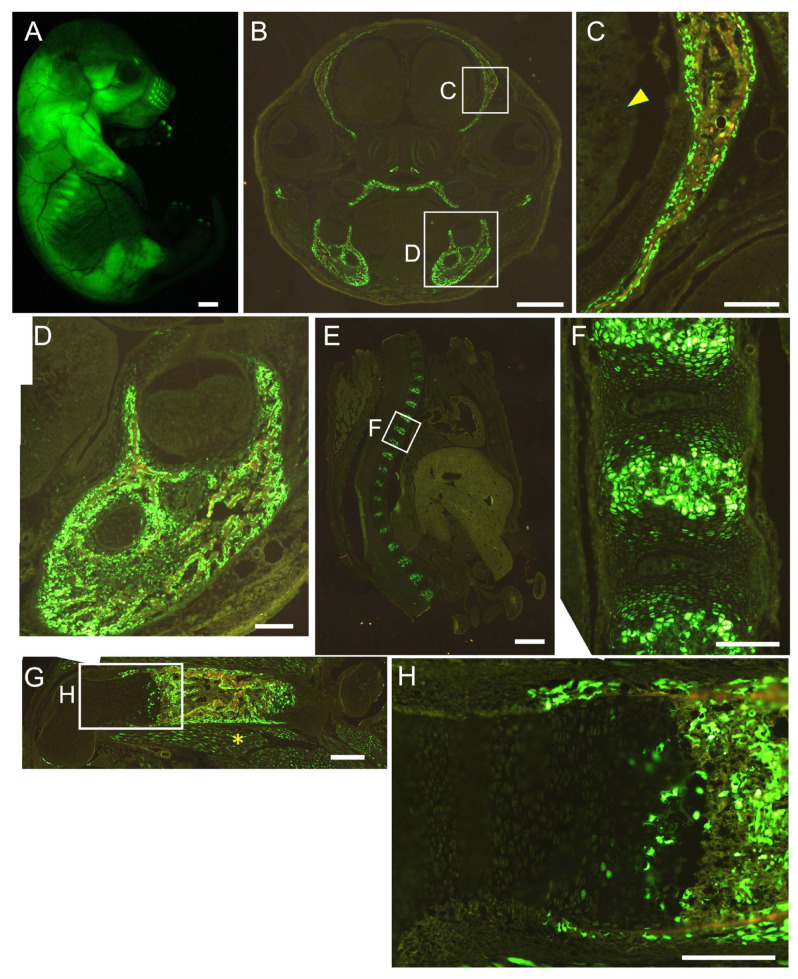
Reporter gene expression of 1.3 kb + 452 × 4-EGFP mice at E16.5. Appearance (**A**) and frozen sections of the head (**B**,**C**), mandible (**D**), vertebrae (**E**,**F**), and femur (**G**,**H**). The boxed regions in (**B**,**E**,**G**) are magnified in (**C**,**D**,**F**,**H**). Arrowhead in (**C**) shows brain, and asterisk in (**G**) shows muscle. Scale bars: 1 mm (**A**,**B**,**E**), 200 μm (**C**,**D**,**F**–**H**). Three out of five F0 transgenic mice analyzed showed a similar expression pattern, and a representative F0 mouse is shown. As the expression pattern of EGFP is partly affected by the integration site of the transgene [34], some transgenic mice showed different expression patterns.

**Figure 6 ijms-26-01653-f006:**
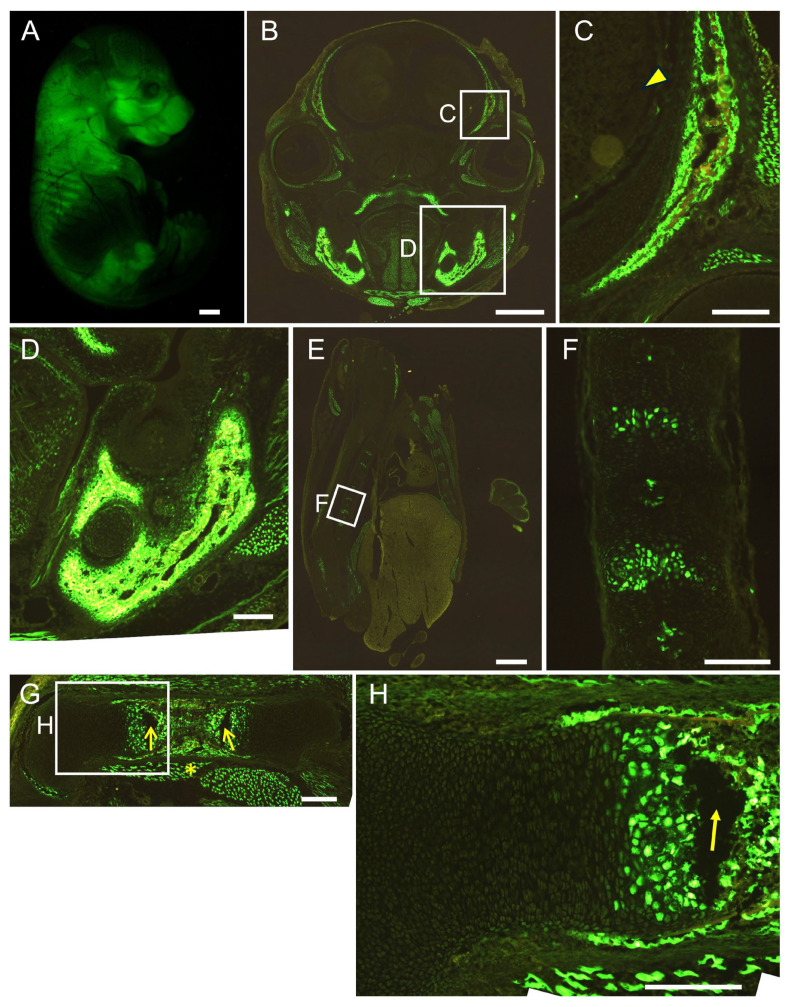
Reporter gene expression of the 1.3 kb + 452 × 4-EGFP mice at E15.5. Appearance (**A**) and frozen section of head (**B**,**C**), mandible (**D**), vertebrae (**E**,**F**), and femur (**G**,**H**). The boxed regions in (**B**,**E**,**G**) are magnified in (**C**,**D**,**F**,**H**). Arrowhead in (**C**) shows brain, and asterisk in (**G**) shows muscle. The empty spaces (arrows in (**G**,**H**)) between the growth plate and bone marrow were generated by artificial breaks in sectioning. Scale bars: 1 mm (**A**,**B**,**E**), 200 μm (**C**,**D**,**F**–**H**). Two out of two F0 transgenic mice analyzed showed a similar expression pattern, and a representative F0 mouse is shown.

**Figure 7 ijms-26-01653-f007:**
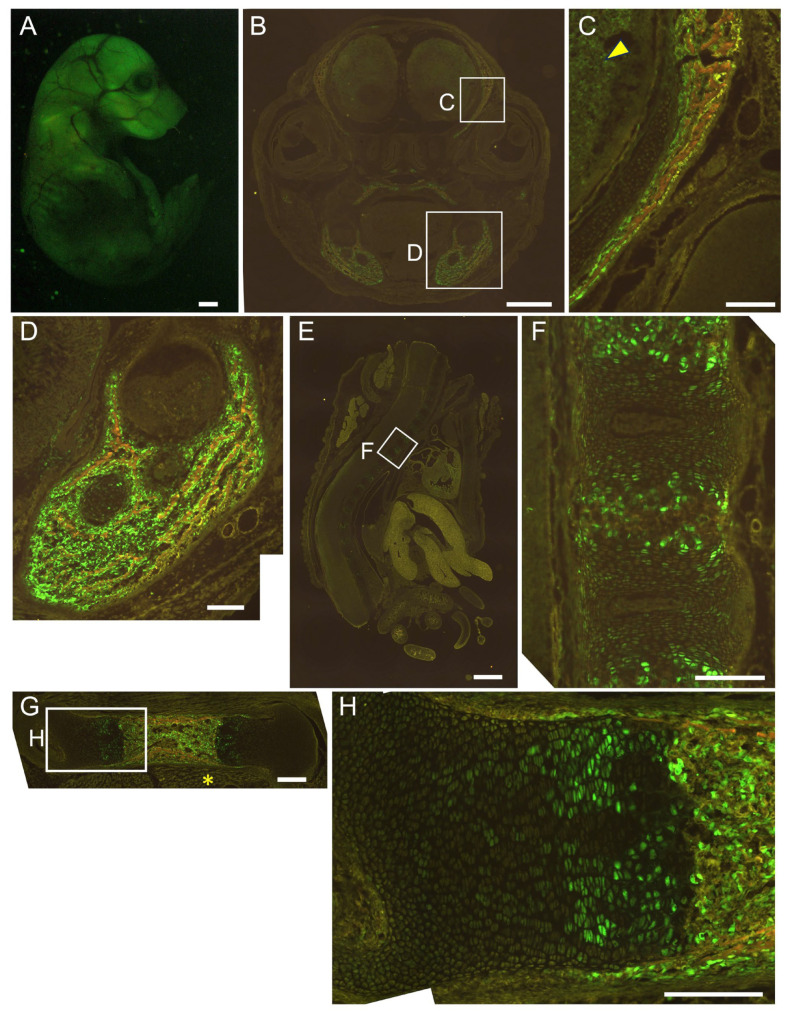
Reporter gene expression of 1.3 kb + 0.8 kb-EGFP mice at E16.5. Appearance (**A**) and frozen sections of the head (**B**,**C**), mandible (**D**), vertebrae (**E**,**F**), and femur (**G**,**H**). The boxed regions in (**B**,**E**,**G**) are magnified in (**C**,**D**,**F**,**H**). Arrowhead in (**C**) shows brain, and asterisk in (**G**) shows muscle. Scale bars: 1 mm (**A**,**B**,**E**), 200 μm (**C**,**D**,**F**–**H**). Three out of four F0 transgenic mice analyzed showed a similar expression pattern, and a representative F0 mouse is shown.

**Figure 8 ijms-26-01653-f008:**
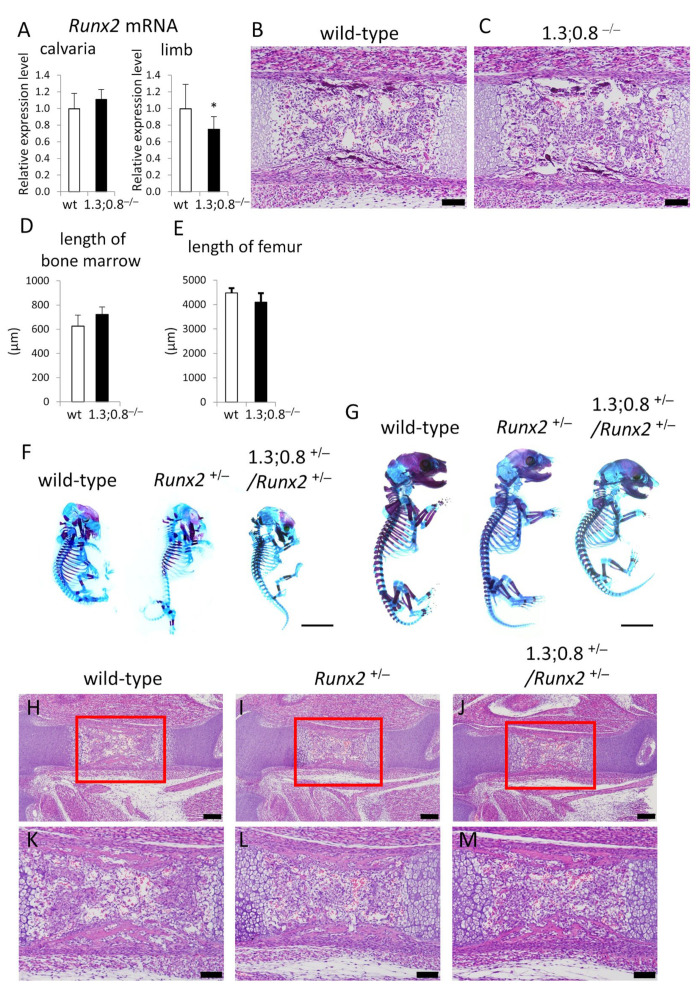
Phenotypes of 1.3 kb- and 0.8 kb-deletion (1.3;0.8^−/−^) mice and 1.3;0.8^+/−^/*Runx2*^+/−^ mice. (**A**–**E**) Analysis of 1.3;0.8^−/−^ mice. *Runx2* mRNA levels in the calvaria and limbs of wild-type and 1.3;0.8^−/−^ newborn mice (**A**), H-E staining of femoral sections of E15.5 embryos (**B**,**C**), the length of bone marrow in femurs at E15.5 (**D**), and the length of femurs at P1 (**E**). (**F**–**M**) Analysis of 1.3;0.8^+/−^/*Runx2*^+/−^ mice. Skeletal preparations of wild-type, *Runx2*^+/−^, and 1.3;0.8^+/−^/*Runx2*^+/−^ embryos at E15.5 (**F**) and E18.5 (**G**), and H-E staining of femoral sections of E15.5 embryos (**H**–**M**). The boxed regions in (**H**–**J**) are magnified in (**K**–**M**), respectively. The number of mice analyzed: (**A**) wild-type, n = 11; 1.3;0.8^−/−^, n = 10; (**D**) n = 3–4; (**E**) n = 3. Skeletal preparations at E15.5 (n = 3–4) and E18.5 (n = 4–6) were examined, and the representatives are shown in (**F**,**G**). Scale bars: 100 μm (**B**,**C**,**K**–**M**), 200 μm (**H**–**J**), 5 mm (**F**,**G**). Data are the means ± SDs. Versus wild type, * *p* < 0.05.

## Data Availability

Data are contained within the article and Appendix A.

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
