# Peer review of "An Osteoblast-Specific Enhancer and Subenhancer Cooperatively Regulate Runx2 Expression in Chondrocytes"

_ijms, 2025, doi:10.3390/ijms26041653_

Round 1
Reviewer 1 Report
Comments and Suggestions for Authors
This manuscript investigates how a 0.8-kb region, located near the 1.3-kb osteoblast-specific enhancer of the Runx2 gene, influences its expression in osteoblasts and chondrocytes in the skeletal development
This manuscript reveals a novel mechanism in the scientific literature, demonstrating how a subenhancer cooperates with an enhancer to regulate tissue-specific expression of the critical transcription factor Runx2 in chondrocytes. Additionally, the study underscores the redundancy and robustness of enhancer networks in developmental genes, contributing valuable insights into the transcriptional regulation of skeletal development and the functional interplay of regulatory elements.
I would like to suggest to include the aims of the manuscript in the last paragraph of the introduction as well as include a paragraph in discussion indicating the limitations of the study and suggestion for future researches
Author Response
Comments 1:
This manuscript investigates how a 0.8-kb region, located near the 1.3-kb osteoblast-specific enhancer of the Runx2 gene, influences its expression in osteoblasts and chondrocytes in the skeletal development
This manuscript reveals a novel mechanism in the scientific literature, demonstrating how a subenhancer cooperates with an enhancer to regulate tissue-specific expression of the critical transcription factor Runx2 in chondrocytes. Additionally, the study underscores the redundancy and robustness of enhancer networks in developmental genes, contributing valuable insights into the transcriptional regulation of skeletal development and the functional interplay of regulatory elements.
I would like to suggest to include the aims of the manuscript in the last paragraph of the introduction as well as include a paragraph in discussion indicating the limitations of the study and suggestion for future researches.
Response 1:
Thank you for this suggestion.
We have added the aims of this study to the last paragraph of the Introduction as follows:
As transcripts from the P1 promoter were detected in both osteoblasts and chondrocytes [1, 2], we searched the regulatory regions for Runx2 expression in chondrocytes in the vicinity of the P1 promoter.
The limitations of the study and suggestions for future research are described in the last paragraph of the Discussion section as follows:
The genomic region examined in this study is restricted to the vicinity of the P1 promoter. As Runx2 is an essential transcription factor for skeletal development, further studies on transcriptional regulation by enhancers distributed in a wide range of the genome are extremely important to obtain a more detailed understanding of the process of skeletal development.
Reviewer 2 Report
Comments and Suggestions for Authors
In this study, the Authors showed cooperation between a Runx2-specific enhancer and neighboring subenhancers. Since Runx2 is an essential transcription factor for osteoblast differentiation and chondrocyte maturation, this study contributes to understanding the spatiotemporal control of this gene expression.
I do not have any comments, and I advise publication as it is.
Author Response
Comments 1:
In this study, the Authors showed cooperation between a Runx2-specific enhancer and neighboring subenhancers. Since Runx2 is an essential transcription factor for osteoblast differentiation and chondrocyte maturation, this study contributes to understanding the spatiotemporal control of this gene expression.
I do not have any comments, and I advise publication as it is.
Response 1:
Thank you for your evaluation.
Reviewer 3 Report
Comments and Suggestions for Authors
The manuscript titled "An osteoblast-specific enhancer and subenhancer cooperatively regulate Runx2 expression in chondrocytes" explores the regulatory elements of Runx2 expression, particularly focusing on the enhancer and subenhancer regions. While the study provides valuable insights, certain aspects require clarification and revision for improved coherence and scientific rigor. Below are specific concerns and suggestions for improvement.
1. Incomplete Discussion of Runx2 Promoters: The authors state that "The transcription of Runx2 is regulated by two promoters, P1 and P2." However, the subsequent discussion focuses only on the P1 promoter, with no details provided on P2. This omission makes the introduction appear incomplete. If P2 is not relevant to this study, it should be explicitly stated to avoid confusion.
2. Unclear Citation and Logical Flow in Enhancer Discussion: The authors mention, "We previously described a 1.3-kb osteoblast-specific Runx2 enhancer that directed reporter EGFP expression specifically to osteoblasts, and showed that 343 bp was sufficient for osteoblast-specific expression." However, this statement lacks a citation. Given that it describes previously published work, a reference should be provided.
Additionally, this sentence appears within a paragraph discussing Runx2 promoters, making the transition unclear. It would be more logical to place this information in a separate paragraph, explicitly linking it to the enhancer discussion rather than the promoter introduction.
3. Definition of EGFP Abbreviation: The abbreviation EGFP appears early in the manuscript but is not explicitly defined. Although commonly known, for clarity and completeness, it should be defined as enhanced green fluorescent protein (EGFP) upon first mention.
4. Lack of Citation for Evolutionary Conservation of Enhancer Region: The authors state, "A 0.8-kb region with homologous sequences has been identified between the 1.3-kb enhancer and P1 promoter in many species." However, no reference is provided for this evolutionary conservation. A supporting citation is necessary to substantiate this claim.
5. Unclear Study Aim: The authors mention, "We examined the enhancer activity of the 0.8-kb region." However, the specific aim of the study is not clearly articulated. The purpose of examining the 0.8-kb region should be explicitly stated in relation to the research hypothesis and broader biological significance.
6. Lack of Proper Citation to Figures: The Discussion section presents interpretations of experimental results but lacks references to specific figures that support those claims. For instance:
The authors state, "Although the single deletion of each of the ten enhancers, which had individual expression patterns with overlap, did not show phenotypes, the deletion of pairs of limb enhancers near the same gene caused phenotypes, indicating that these enhancers function redundantly in limb development."
Where can this result be found? Which figure supports this claim?
Similarly, the authors state, "Although the histone of the 0.8-kb region was modified similar to an enhancer, 0.8-kb DNA fragment did not exhibit enhancer activity in vitro."
Where is this result documented? It should be explicitly linked to a figure.
Proper citation of figures within the Discussion would significantly improve clarity and allow readers to verify claims more easily.
7. Inconsistent Use of Terminology: The authors interchangeably use subenhancer and enhancer-like region when referring to the 0.8-kb conserved region. However, the manuscript does not clearly define what distinguishes a subenhancer from a typical enhancer.
In some cases, the term shadow enhancer is mentioned in the discussion, but it is not explicitly stated whether the 0.8-kb region qualifies as a shadow enhancer or whether it has distinct characteristics. A clearer definition and classification would help avoid ambiguity.
8. Logical Inconsistency Regarding the Function of the 0.8-kb Region: The authors claim that the 0.8-kb region alone lacks enhancer activity ("Although the histone of the 0.8-kb region was modified similar to an enhancer, 0.8-kb DNA fragment did not exhibit enhancer activity in vitro."). However, they later state that when combined with the 1.3-kb enhancer, the 0.8-kb region functions as a subenhancer to induce Runx2 expression in chondrocytes. This raises the question: What molecular mechanism allows the 0.8-kb region to function only in combination with the 1.3-kb enhancer? If the 0.8-kb region has no independent enhancer activity, could its role simply be to facilitate chromatin accessibility rather than act as a bona fide regulatory element? The manuscript should address this mechanistic gap.
9. Unclear Experimental Design in Luciferase Assays: The study tests the enhancer activity of the 0.8-kb region in a luciferase assay (Figure 2C). However: The assay only measures transcriptional activation and does not assess chromatin conformation changes, which might be relevant for a subenhancer. The authors test the 0.8-kb region alone but do not provide data on whether the 1.3-kb enhancer influences its activity in a luciferase system. They state, "The 452×4 vector strongly induced luciferase activity in human chondrocyte cell lines, but not in the human osteoblast cell line SaOS2." However, this does not necessarily mean the region is chondrocyte-specific—it could be due to differences in chromatin accessibility or transcription factor availability. Control experiments using a known osteoblast-specific enhancer would strengthen these claims.
10. Contradictory Findings Regarding In Vivo vs. In Vitro Enhancer Activity: In luciferase assays, the 0.8-kb region does not show strong enhancer activity. However, in in vivo EGFP reporter mice, it contributes to chondrocyte-specific expression.
This discrepancy is not well explained. The authors need to discuss: Whether chromatin remodeling or additional transcription factors are necessary in vivo.
Whether the experimental conditions used in vitro (e.g., cell lines, culture medium) differ from the in vivo microenvironment.
Whether the Runx2 regulatory landscape includes higher-order chromatin interactions that cannot be replicated in a luciferase assay.
11. Weak Statistical Justification: The manuscript lacks clear descriptions of statistical tests for some data sets.
For example, in the micro-CT analysis (Figure 1E), statistical comparisons are not described in sufficient detail. Are these comparisons based on a t-test, ANOVA, or another statistical method? Were normality and variance homogeneity tested?
The statement, "Three out of five transgenic mice analyzed showed a similar expression pattern," suggests variability in the results, but no explanation is given regarding why some mice exhibited different patterns.
Sample sizes for some figures (e.g., n=3-4 in certain skeletal preparations) may be too small to draw robust conclusions. The authors should provide power analyses or justification for their sample sizes.
12. Incomplete Discussion of Runx2 Regulation in Chondrocytes: The introduction and discussion primarily focus on osteoblast-specific regulation of Runx2 and only briefly mention chondrocytes. However: Runx2 has distinct regulatory mechanisms in chondrocytes compared to osteoblasts.
The authors do not reference well-established studies on Runx2 regulation in chondrocytes, including key pathways such as SOX9-mediated repression or BMP signaling.
If the Runx2 enhancer plays a role in chondrocyte expression, how does it interact with known chondrocyte-specific regulators? This should be addressed.
13. Lack of Consideration for Alternative Explanations: The authors conclude that the 0.8-kb region functions as a subenhancer, but other interpretations exist:
It could act as a chromatin scaffold rather than a transcriptional enhancer.
It might serve as a boundary element that helps position the 1.3-kb enhancer within the correct chromatin context.
Its regulatory function may depend on enhancer-promoter looping interactions rather than direct activation.
The manuscript does not address these possibilities, which would provide a more balanced discussion.
Author Response
Thank you for the constructive suggestions.
The manuscript titled "An osteoblast-specific enhancer and subenhancer cooperatively regulate Runx2 expression in chondrocytes" explores the regulatory elements of Runx2 expression, particularly focusing on the enhancer and subenhancer regions. While the study provides valuable insights, certain aspects require clarification and revision for improved coherence and scientific rigor. Below are specific concerns and suggestions for improvement.
Comments 1: Incomplete Discussion of Runx2 Promoters: The authors state that "The transcription of Runx2 is regulated by two promoters, P1 and P2." However, the subsequent discussion focuses only on the P1 promoter, with no details provided on P2. This omission makes the introduction appear incomplete. If P2 is not relevant to this study, it should be explicitly stated to avoid confusion.
Response 1: P2 promoter has not been well studied. Transcripts from the P2 promoter were detected not only in osteoblasts but also in osteoprogenitors. As the 1.3 kb osteoblast-specific enhancer directed reporter gene expression to osteoblasts but not to osteoprogenitors, the 1.3 kb enhancer is likely to interact with the P1 promoter. These descriptions have been added to the Introduction to avoid confusion.
Comments 2-1: Unclear Citation and Logical Flow in Enhancer Discussion: The authors mention, "We previously described a 1.3-kb osteoblast-specific Runx2 enhancer that directed reporter EGFP expression specifically to osteoblasts, and showed that 343 bp was sufficient for osteoblast-specific expression." However, this statement lacks a citation. Given that it describes previously published work, a reference should be provided.
Response 2-1: References have been added.
Comments 2-2: Additionally, this sentence appears within a paragraph discussing Runx2 promoters, making the transition unclear. It would be more logical to place this information in a separate paragraph, explicitly linking it to the enhancer discussion rather than the promoter introduction.
Response 2-2: A description of the enhancer has been placed in a separate paragraph.
Comments 3: Definition of EGFP Abbreviation: The abbreviation EGFP appears early in the manuscript but is not explicitly defined. Although commonly known, for clarity and completeness, it should be defined as enhanced green fluorescent protein (EGFP) upon first mention.
Response 3: EGFP was defined as enhanced green fluorescent protein.
Comments 4: Lack of Citation for Evolutionary Conservation of Enhancer Region: The authors state, "A 0.8-kb region with homologous sequences has been identified between the 1.3-kb enhancer and P1 promoter in many species." However, no reference is provided for this evolutionary conservation. A supporting citation is necessary to substantiate this claim.
Response 4: The data were derived from the UCSC Genome Browser on Mouse, July 2007, NCBI/mm9, chr17:44984700-44949500, and are described in the last paragraph of the Introduction.
Comments 5: Unclear Study Aim: The authors mention, "We examined the enhancer activity of the 0.8-kb region." However, the specific aim of the study is not clearly articulated. The purpose of examining the 0.8-kb region should be explicitly stated in relation to the research hypothesis and broader biological significance.
Response 5: As transcripts from the P1 promoter were detected in both osteoblasts and chondrocytes [1, 2], we searched the regulatory regions for Runx2 expression in chondrocytes in the vicinity of the P1 promoter. This has been described in the last paragraph of the Introduction.
Comments 6: Lack of Proper Citation to Figures: The Discussion section presents interpretations of experimental results but lacks references to specific figures that support those claims. For instance:
The authors state, "Although the single deletion of each of the ten enhancers, which had individual expression patterns with overlap, did not show phenotypes, the deletion of pairs of limb enhancers near the same gene caused phenotypes, indicating that these enhancers function redundantly in limb development."
Where can this result be found? Which figure supports this claim?
Similarly, the authors state, "Although the histone of the 0.8-kb region was modified similar to an enhancer, 0.8-kb DNA fragment did not exhibit enhancer activity in vitro."
Where is this result documented? It should be explicitly linked to a figure.
Proper citation of figures within the Discussion would significantly improve clarity and allow readers to verify claims more easily.
Response 6: We have cited these figures in the Discussion section. Further, we have changed the order of the discussion (paragraph 1 was moved to paragraph 2) to avoid confusion.
Comments 7-1: Inconsistent Use of Terminology: The authors interchangeably use subenhancer and enhancer-like region when referring to the 0.8-kb conserved region. However, the manuscript does not clearly define what distinguishes a subenhancer from a typical enhancer.
Response 7-1: We clearly defined the enhancer and subenhancer. This is described in the first paragraph of the Discussion section as follows.
Therefore, the 0.8-kb region was not an enhancer, which regulates the spatiotemporal expression of the corresponding gene, and appeared to be a subenhancer that does not exhibit sufficient activity as an enhancer, but acquires enhancer activity when combined with an enhancer.
Comments 7-2: In some cases, the term shadow enhancer is mentioned in the discussion, but it is not explicitly stated whether the 0.8-kb region qualifies as a shadow enhancer or whether it has distinct characteristics. A clearer definition and classification would help avoid ambiguity.
Response 7-2: The difference between shadow enhancers and the 0.8-kb region is described in the last part of the second paragraph of the Discussion as follows:
These shadow or redundant enhancers in Drosophila and mice have individual expression patterns with overlap and function additively to ensure phenotypic robustness or faithful expression patterns. However, the 0.8-kb region alone directed EGFP expression to neither osteoblasts nor chondrocytes, in which Runx2 is expressed. Furthermore, combined with the 1.3-kb osteoblast-specific enhancer, the 0.8-kb region did not function additively to induce expression in osteoblasts, but induced expression in a different lineage of cells, chondrocytes (Figs. 4-7). Therefore, the 0.8-kb region is not a shadow or redundant enhancer.
Comments 8: Logical Inconsistency Regarding the Function of the 0.8-kb Region: The authors claim that the 0.8-kb region alone lacks enhancer activity ("Although the histone of the 0.8-kb region was modified similar to an enhancer, 0.8-kb DNA fragment did not exhibit enhancer activity in vitro."). However, they later state that when combined with the 1.3-kb enhancer, the 0.8-kb region functions as a subenhancer to induce Runx2 expression in chondrocytes. This raises the question: What molecular mechanism allows the 0.8-kb region to function only in combination with the 1.3-kb enhancer? If the 0.8-kb region has no independent enhancer activity, could its role simply be to facilitate chromatin accessibility rather than act as a bona fide regulatory element? The manuscript should address this mechanistic gap.
Response 8: We generated figure 9 to explain the proposed mechanism of Runx2 transcription in chondrocytes by an enhancer and a subenhancer. We propose that the 0.8-kb subenhancer alone cannot interact with the promoter but the 1.3-kb osteoblast-specific enhancer and the 0.8-kb subenhancer cooperatively interact with the promoter and induce Runx2 transcription in chondrocytes (Fig. 9). The 452x4 luciferase vector induced activity in chondrocyte cell lines, whereas the 452x4 DNA fragment failed to induce EGFP expression in chondrocytes in vivo (Figs. 2E and 4). Therefore, it is speculated that the chromatin modifiers in the 1.3-kb osteoblast-specific enhancer complex enhanced chromatin modification in the 0.8-kb subenhancer and/or the 1.3-kb enhancer complex supplied the transcription factors and cofactors to the 0.8-kb subenhancer, both of which led to the interaction of three complexes (enhancer, subenhancer, and promoter) in chondrocytes in vivo (Fig. 9). As the chromatin of the 0.8-kb region was modified as an enhancer in a chondrogenic cell line and primary chondrocytes (Fig. 2A, B), the 0.8-kb region was considered to be activated, as shown in Fig. 9C.
Comments 9: Unclear Experimental Design in Luciferase Assays: The study tests the enhancer activity of the 0.8-kb region in a luciferase assay (Figure 2C). However: The assay only measures transcriptional activation and does not assess chromatin conformation changes, which might be relevant for a subenhancer. The authors test the 0.8-kb region alone but do not provide data on whether the 1.3-kb enhancer influences its activity in a luciferase system. They state, "The 452×4 vector strongly induced luciferase activity in human chondrocyte cell lines, but not in the human osteoblast cell line SaOS2." However, this does not necessarily mean the region is chondrocyte-specific—it could be due to differences in chromatin accessibility or transcription factor availability. Control experiments using a known osteoblast-specific enhancer would strengthen these claims.
Response 9: In the luciferase assays, the luciferase vector containing the 0.8-kb DNA fragment and minimal promoter was transiently transfected into the cell lines to evaluate the enhancer activity of the 0.8-kb DNA (Fig. 2C, Supplementary figure S2B). As the luciferase vector is not integrated into the genome by transient transfection, chromatin conformational changes cannot be assessed. In Fig. 2E and F, the vector containing the core region of 1.3 kb (343x2) and core region of 0.8 kb (452x4) were generated, and luciferase assay was performed. The combined vector showed luciferase activity similar to that of 452×4 in the chondrocyte cell lines, but failed to induce luciferase activity in the osteoblast cell line. As previously reported, the core region of 1.3 kb (343 bp) is osteoblast-specific [3]. We used 343x2 as an osteoblast-specific enhancer (Fig. 2E, F).
Comments 10: Contradictory Findings Regarding In Vivo vs. In Vitro Enhancer Activity: In luciferase assays, the 0.8-kb region does not show strong enhancer activity. However, in in vivo EGFP reporter mice, it contributes to chondrocyte-specific expression.
This discrepancy is not well explained. The authors need to discuss: Whether chromatin remodeling or additional transcription factors are necessary in vivo.
Whether the experimental conditions used in vitro (e.g., cell lines, culture medium) differ from the in vivo microenvironment.
Whether the Runx2 regulatory landscape includes higher-order chromatin interactions that cannot be replicated in a luciferase assay.
Response 10: We have added Figure 9 to explain the mechanism of subenhancer-mediated transcription in chondrocytes in vivo. We have discussed this in the last part of the second paragraph of the Discussion section as follows:
We propose that the 0.8-kb subenhancer alone cannot interact with the promoter but the 1.3-kb osteoblast-specific enhancer and the 0.8-kb subenhancer cooperatively interact with the promoter and induce Runx2 transcription in chondrocytes (Fig. 9). The 452x4 luciferase vector induced activity in chondrocyte cell lines, whereas the 452x4 DNA fragment failed to induce EGFP expression in chondrocytes in vivo (Figs. 2E and 4). Therefore, it is speculated that the chromatin modifiers in the 1.3-kb osteoblast-specific enhancer complex enhanced chromatin modification in the 0.8-kb subenhancer and/or the 1.3-kb enhancer complex supplied the transcription factors and cofactors to the 0.8-kb subenhancer, either of which led to the interaction of three complexes (enhancer, subenhancer, and promoter) in chondrocytes in vivo (Fig. 9).
Comments 11-1: Weak Statistical Justification: The manuscript lacks clear descriptions of statistical tests for some data sets.
For example, in the micro-CT analysis (Figure 1E), statistical comparisons are not described in sufficient detail. Are these comparisons based on a t-test, ANOVA, or another statistical method? Were normality and variance homogeneity tested?
Response 11-1: Normality and homogeneity of variance were tested, and statistical analyses of two groups were performed using Student’s t-test and those of more than three groups by one-way ANOVA. In Fig. 1E, statistical analysis was performed using a t-test. We added “Normality and homogeneity of variance were tested” in the method of statistical analysis.
Comments 11-2: The statement, "Three out of five transgenic mice analyzed showed a similar expression pattern," suggests variability in the results, but no explanation is given regarding why some mice exhibited different patterns.
Response 11-2: We have explained this in the legend of Figure 5 as follows.
As the expression pattern of EGFP is partly affected by the integration site of the transgene [4], some transgenic mice showed different expression patterns.
Comments 11-3: Sample sizes for some figures (e.g., n=3-4 in certain skeletal preparations) may be too small to draw robust conclusions. The authors should provide power analyses or justification for their sample sizes.
Response 11-3: We think that the sample size of skeletal preparation at E15.5 (n=3-4) was sufficient, because we also analyzed histology at E16.5 (n=6-7) and skeletal preparation at E18.5 (n=4-6) comparing Runx2+/– and 1.3;0.8+/–/Runx2+/– mice and obtained consistent results.
Comments 12-1: Incomplete Discussion of Runx2 Regulation in Chondrocytes: The introduction and discussion primarily focus on osteoblast-specific regulation of Runx2 and only briefly mention chondrocytes. However: Runx2 has distinct regulatory mechanisms in chondrocytes compared to osteoblasts.
The authors do not reference well-established studies on Runx2 regulation in chondrocytes, including key pathways such as SOX9-mediated repression or BMP signaling.
Response 12-1: We have added the references of the studies on Runx2 regulation in chondrocytes in the Introduction as follows:
Bone morphogenetic protein (BMP) and Wnt signaling pathways induce Runx2 expression and chondrocyte maturation [5-7]. Further, Sox9 suppresses Runx2 expression and prevents chondrocyte maturation through the induction of Nkx3-2 expression [8, 9].
Comments 12-2: If the Runx2 enhancer plays a role in chondrocyte expression, how does it interact with known chondrocyte-specific regulators? This should be addressed.
Response 12-2: We generated Figure 9 and explained how the 1.3-kb enhancer and 0.8-kb subenhancer interact with transcription factors and cofactors.
Comments 13-1: Lack of Consideration for Alternative Explanations: The authors conclude that the 0.8-kb region functions as a subenhancer, but other interpretations exist:
It might serve as a boundary element that helps position the 1.3-kb enhancer within the correct chromatin context.
Response 13-1: As the 1.3-kb enhancer alone functions as an osteoblast-specific enhancer, we believe it is unlikely.
Comments 13-2: It could act as a chromatin scaffold rather than a transcriptional enhancer.
Its regulatory function may depend on enhancer-promoter looping interactions rather than direct activation.
Response 13-2: As the 1.3 kb, 452x4 or 0.8 kb, and Hsp68 minimal promoter were inserted in juxtaposition in the transgenic mice without intervening genomic DNA, we think that the 0.8-kb region is unlikely to function as chromatin scaffold, and it is also unlikely that the function of the 0.8-kb region depends on enhancer-promoter looping interaction.
Comments 13-3: The manuscript does not address these possibilities, which would provide a more balanced discussion.
Response 13-3: We generated Figure 9 and discussed the possible mechanism of the functions of the 0.8-kb region.
- Enomoto, H.; Enomoto-Iwamoto, M.; Iwamoto, M.; Nomura, S.; Himeno, M.; Kitamura, Y.; Kishimoto, T.; Komori, T., Cbfa1 is a positive regulatory factor in chondrocyte maturation. J Biol Chem 2000, 275, (12), 8695-702.
- Liu, J. C.; Lengner, C. J.; Gaur, T.; Lou, Y.; Hussain, S.; Jones, M. D.; Borodic, B.; Colby, J. L.; Steinman, H. A.; van Wijnen, A. J.; Stein, J. L.; Jones, S. N.; Stein, G. S.; Lian, J. B., Runx2 protein expression utilizes the Runx2 P1 promoter to establish osteoprogenitor cell number for normal bone formation. J Biol Chem 2011, 286, (34), 30057-70.
- Kawane, T.; Komori, H.; Liu, W.; Moriishi, T.; Miyazaki, T.; Mori, M.; Matsuo, Y.; Takada, Y.; Izumi, S.; Jiang, Q.; Nishimura, R.; Kawai, Y.; Komori, T., Dlx5 and mef2 regulate a novel runx2 enhancer for osteoblast-specific expression. J Bone Miner Res 2014, 29, (9), 1960-9.
- Tasic, B.; Hippenmeyer, S.; Wang, C.; Gamboa, M.; Zong, H.; Chen-Tsai, Y.; Luo, L., Site-specific integrase-mediated transgenesis in mice via pronuclear injection. Proceedings of the National Academy of Sciences of the United States of America 2011, 108, (19), 7902-7.
- Shu, B.; Zhang, M.; Xie, R.; Wang, M.; Jin, H.; Hou, W.; Tang, D.; Harris, S. E.; Mishina, Y.; O'Keefe, R. J.; Hilton, M. J.; Wang, Y.; Chen, D., BMP2, but not BMP4, is crucial for chondrocyte proliferation and maturation during endochondral bone development. J Cell Sci 2011, 124, (Pt 20), 3428-40.
- Yan, J.; Li, J.; Hu, J.; Zhang, L.; Wei, C.; Sultana, N.; Cai, X.; Zhang, W.; Cai, C. L., Smad4 deficiency impairs chondrocyte hypertrophy via the Runx2 transcription factor in mouse skeletal development. J Biol Chem 2018, 293, (24), 9162-9175.
- Dong, Y. F.; Soung do, Y.; Schwarz, E. M.; O'Keefe, R. J.; Drissi, H., Wnt induction of chondrocyte hypertrophy through the Runx2 transcription factor. J Cell Physiol 2006, 208, (1), 77-86.
- Lengner, C. J.; Hassan, M. Q.; Serra, R. W.; Lepper, C.; van Wijnen, A. J.; Stein, J. L.; Lian, J. B.; Stein, G. S., Nkx3.2-mediated repression of Runx2 promotes chondrogenic differentiation. J Biol Chem 2005, 280, (16), 15872-9.
- Yamashita, S.; Andoh, M.; Ueno-Kudoh, H.; Sato, T.; Miyaki, S.; Asahara, H., Sox9 directly promotes Bapx1 gene expression to repress Runx2 in chondrocytes. Experimental cell research 2009, 315, (13), 2231-40.
Round 2
Reviewer 3 Report
Comments and Suggestions for Authors
Authors addressed all issues raised by reviewer properly. I don't have further comment on this article.